# Multidimensional Correlates of Parental Self-Efficacy in Managing Adolescent Internet Use among Parents of Adolescents with Attention-Deficit/Hyperactivity Disorder

**DOI:** 10.3390/ijerph17165768

**Published:** 2020-08-10

**Authors:** Yi-Ping Hsieh, Chia-Fen Wu, Wen-Jiun Chou, Cheng-Fang Yen

**Affiliations:** 1Department of Social Work, College of Nursing and Professional Disciplines, University of North Dakota, Grand Forks, ND 58202, USA; yiping66@gmail.com; 2Department of Psychiatry, Kaohsiung Medical University Hospital, Kaohsiung 80708, Taiwan; pino3015@hotmail.com; 3College of Medicine, Chang Gung University, Taoyuan 33302, Taiwan; 4Department of Child and Adolescent Psychiatry, Chang Gung Memorial Hospital, Kaohsiung Medical Center, Kaohsiung 83301, Taiwan; 5Department of Psychiatry, School of Medicine, Kaohsiung Medical University, Kaohsiung 80708, Taiwan

**Keywords:** parental self-efficacy, ADHD, Internet addiction, parenting, depressive symptoms, adolescents’ Internet use

## Abstract

Given the growing concerns of problematic Internet use and online safety, it is critical to address parental self-efficacy in managing adolescent Internet use and to examine associated factors, especially in parents of adolescents with attention-deficit/hyperactivity disorder (ADHD). We examined the roles of adolescents’ hyperactivity/impulsivity, inattention and oppositional defiant disorder (ODD) symptoms, parents’ depressive symptoms, parenting behavior (parental care and indifference), and child behavior (Internet addiction) in relation to parental self-efficacy in managing adolescent Internet use. We recruited 237 Taiwanese parents of adolescents with ADHD (ages 11–18). Hierarchical linear regression was performed in four steps to test the study hypotheses. The results indicated that child’s age, ODD symptoms, and Internet addiction of adolescents were negatively associated, and parental care was positively associated with parental self-efficacy in managing adolescent Internet use. The final model was significant and explained 43% of the variance. The present study demonstrated that parenting and child behaviors contribute to parental self-efficacy in managing adolescent Internet use. Moreover, children’s ODD symptoms were identified as the risk factor for reduced parental self-efficacy.

## 1. Introduction

### 1.1. Parental Self-Efficacy in Managing Adolescent Internet Use

Self-efficacy refers to an individual’s belief in their capacity to perform behaviors necessary to produce specific performance attainments [1,2,3]. Parental self-efficacy refers to a parent’s belief in their competence to perform the parenting role successfully and foster their children’s positive development and adjustment [4]. Higher levels of parental self-efficacy are related to more positive parenting practices and adaptive/nurturing child-rearing environment, which in turn leads to lower levels of adolescents’ externalizing problems [5] and overall parent and child well-being [4]. Parental self-efficacy is also effective in reducing problematic child behaviors [4]. Glidewell and Livert described parental self-efficacy as situation-specific according to the task and context [6]. In this digital age, individuals can do almost anything online, such as learning, searching, posting, shopping, social networking, finding entertainment, gaming, and dating. However, Internet use can also have side effects, such as Internet addiction, harassment, and cybercrime. Internet use is convenient, highly accessible, affordable, and attractive, which makes adolescents spend many hours online [7]. Problematic Internet use and online safety have become one of the major concerns in parenting. Thus, it is critical to examine and address parental self-efficacy and associated factors in managing adolescent Internet use. The present study examined the roles of adolescent and parental mental health, parenting behavior, and adolescent behavior in relation to parental self-efficacy in managing adolescent Internet use.

### 1.2. Mental Health and Parental Self-Efficacy

Attention-deficit/hyperactivity disorder (ADHD) is one of the most prevalent DSM-5 mental disorders among children and adolescents in Taiwan (8.7%) [8]. Children’s ADHD symptoms have been linked to lower parental self-efficacy levels [9,10]. Children with ADHD are more likely than those without ADHD to have temper outbursts, low self-regulation, and low frustration tolerance and tend to be less able to follow parents’ instructions and house rules [11,12]. In response to the child-rearing challenges, parents of children with ADHD may use more punitive disciplinary strategies, express more disapproval [13], perceive increased parenting stress, and experience decreased parental self-efficacy [11,14,15].

The Swanson, Nolan, and Pelham version IV Scale (SNAP-IV) is a widely used rating scale that measures the ADHD symptoms of hyperactivity/impulsivity and inattention and oppositional defiant disorder (ODD) symptoms. Hyperactivity/impulsivity is characterized by symptoms of impulsivity and hyperactivity. Inattention is characterized by distractibility, limited attention span, forgetfulness, and procrastination. ADHD children with inattention symptoms have difficulty in following detailed instructions, remaining focused, and organizing tasks. ODD is a common disorder in children marked by hostile and defiant behavior [16]. Parents of children with significant ODD symptoms tend to view their children as more difficult to manage and view themselves as incompetent and overwhelmed [17]. Thus, we hypothesized that higher levels of ADHD and ODD symptoms are associated with lower parental self-efficacy levels in managing ADHD children’s Internet use.

In addition to children’s mental health, parental depression has been negatively associated with parental self-efficacy [18] and positively with children’s Internet addiction [19].

### 1.3. Parenting Behavior and Parental Self-Efficacy

Parents who reported more positive parent–adolescent communication perceived higher parental self-efficacy than did those who reported less positive communication [5]. Parental care and parental indifference are two parenting behaviors measured in the widely used Parental Bonding Instrument (PBI) [20,21]. Parental care represents warm and caring parenting behavior, whereas parental indifference represents cold and intrusive parenting behavior [22]. Fathers’ self-efficacy is associated with warm parental behavior in Asian society [23]. Similarly, mothers who perceive high parental self-efficacy are more likely to display warm and caring parenting behaviors toward their children and adopt an authoritative parenting style [24].

By contrast, parental indifference to children’s emotions may be considered a form of emotional invalidation [25] as well as psychological neglect and parental rejection. Parental indifference was associated with negative outcomes among children, such as psychological maladjustment and negative personality dispositions, across ethnicities, cultures, and geographical boundaries [26]. Parental indifference and rejection were associated with increased risks of adolescents’ problematic Internet use (e.g., Internet gaming disorder, software piracy, and illegal downloading of copyrighted media files) through low core self-evaluations and low self-control [27,28]. Parental self-efficacy is strongly influenced by performance accomplishments [2] and may therefore be related to parents’ actual experiences with their children (either failures or successes) in parenting. When parents have a difficult time disciplining and communicating with their children, parents may feel frustrated, exhibit indifference toward the child, and perceive low parental self-efficacy.

### 1.4. Child Behavior and Parental Self-Efficacy

Previous research has documented that parental behaviors play a vital role in predicting children’s and adolescents’ psychological and behavioral adjustment [29]. For example, Hsieh and coworkers found that parents’ physical and psychological neglect and authoritarian parenting and shaming were positively associated with children’s Internet addiction, whereas authoritative parenting and positive parent–child relationships were negatively associated with children’s Internet addiction [30,31]. Although complex associations between parents’ and children’s behaviors were mostly studied in parental influence and view children as passive recipients of parental behaviors [32], children may actively influence parental behavior and self-efficacy. Thus, it is essential to explore how child behavior or condition may influence parental self-efficacy and parenting. Only a few studies have examined child-driven processes in the relationship between parenting and child adjustment. For example, in response to children’s antisocial behaviors, parents adopt increased negative harsh control and decreased positive parenting behaviors [33]. When children exhibit disruptive behaviors, their parents’ behaviors and self-efficacy are more likely to be influenced. A longitudinal study revealed that child externalizing behavior predicted negative parental control through parental self-efficacy [34]. High levels of children’s disruptive behaviors may make parents feel more frustrated and less confident in parenting and experience lower parental self-efficacy levels. This child-driven processes may also apply to the relationship between adolescents’ Internet addiction and parental self-efficacy in managing adolescents’ Internet use, although no study has been conducted to test this hypothesis. In other words, parents who have adolescents with higher levels of Internet addiction may be more likely to perceive lower parental self-efficacy in managing adolescents’ Internet use.

### 1.5. Aims of the Present Study

Given the growing concerns of adolescents’ problematic Internet use, it is vital to examine and address parental self-efficacy in managing it and factors associated with it. The present study examined the roles of adolescents’ and parents’ mental health (ADHD and ODD symptoms in adolescents and parents’ depressive symptoms), parenting behavior (parental care and indifference), and adolescent behavior (Internet addiction) in relation to parental self-efficacy in managing adolescent Internet use after controlling for demographic factors (children’s sex and age and parental education). We hypothesized that higher levels of ADHD and ODD symptoms, parents’ depressive symptom, parental indifference, and adolescent Internet addiction are associated with lower parental self-efficacy levels in managing adolescent Internet use, whereas parental care is associated with higher parental self-efficacy levels in managing adolescent Internet use.

## 2. Methods

### 2.1. Participants and Procedure

Parents of adolescents aged 11–18 years who had been diagnosed as having ADHD, according to the Diagnostic and Statistical Manual of Mental Disorders, Fifth Edition (DSM-5) [16], were consecutively recruited into this study between August 2014 and July 2015 from the child and adolescent psychiatric outpatient clinics of two medical centers in Kaohsiung, Taiwan. Two child psychiatrists conducted diagnostic interviews with the parents to establish the diagnosis of ADHD according to the DSM-5 criteria. Multiple data sources, including clinical observation of the adolescents’ behavior and the parents’ rating of ADHD symptoms on the short version of SNAP-IV–Chinese version [35,36], were also used to support the diagnosis. Parents of adolescents with intellectual disability, schizophrenia, bipolar disorder, autistic disorder with difficulties in communication, or any cognitive deficits that resulted in significant behavioral and emotional difficulties were excluded.

In total, 237 parents of adolescents with ADHD were invited; of them, 231 (97.5%) agreed to participate (83% mothers). In all, 86% of the adolescents with ADHD were male, and 82% of participants were in a two-parent family. Only one parent per household answered the questionnaire. Informed consent was obtained from all the participants prior to the assessment. The participants could seek help from the research assistants if they had problems in completing the questionnaires.

### 2.2. Instruments

#### 2.2.1. Parental Self-Efficacy in Managing Adolescent Internet Use (PSMIS)

We used the PSMIS [37] to measure parents’ self-efficacy in managing adolescent Internet use in the previous month. The scale contained 18 items in four dimensions: safety management (6 items), parental reasoning (4 items), rule-setting practices (4 items), and parental monitoring (4 items). Each item was rated on a 7-point Likert scale ranging from 0 (no efficacy at all) to 6 (extremely strong efficacy). The reliability and validity of the PSMIS have been described elsewhere [37]. The mean scores were calculated. A higher score indicates a higher level of parental self-efficacy to manage adolescent Internet use. The Cronbach’s α of the PSMIS in the present study was 0.96, which demonstrated strong internal consistency.

#### 2.2.2. The Swanson, Nolan, and Pelham Version IV Scale—Chinese Version (SNAP-IV)

The 26-item SNAP-IV—Chinese version was used for assessing the severities of DSM-IV-derived ADHD and ODD symptoms in the previous month [35,36]. Each item is rated on a 4-point Likert scale ranging from 0 (not at all) to 3 (very much). The mean scores were calculated. The Cronbach’s α of inattention, hyperactivity/impulsivity, and oppositional defiant subscales in the present study were 0.91, 0.90, and 0.93, respectively, indicating excellent internal consistency.

#### 2.2.3. Mandarin Chinese Version of the Center for Epidemiologic Studies Depression Scale (CES-D)

The Mandarin Chinese version [38] of the CES-D [39] comprises 20 items that assess the frequency of parental depressive symptoms in the previous month. Each item is rated on a 4-point Likert scale ranging from 0 (never) to 3 (always). The mean scores were calculated. A higher score indicates more severe depression. The Cronbach’s α in the present study was 0.81, indicating good internal consistency.

#### 2.2.4. Parental Bonding Instrument (PBI)

We used the Chinese version of the PBI [20,21,40] to measure parenting behavior in two dimensions: parental care (6 items) and parental indifference (6 items). Sample questions for care include “spoke to the child in a warm and friendly voice”, “was affectionate to the child”, and “enjoyed talking things over with the child”. Sample questions for indifference include “did not seem to understand what the child needed or wanted”, “did not talk with me very much”, and “seemed emotionally cold to the child”. Each item is rated on a 4-point Likert scale ranging from 1 (very likely) to 4 (very unlikely). We reverse-coded the items to make the interpretation easier. A high score on the parental care dimension reflects more affection and warmth, whereas a high score on the parental indifference dimension indicates more indifference or rejection. Mean scores were calculated for each subscale. The reliability and validity of the Chinese PBI have been described elsewhere [20]. The Cronbach’s α of parent-reported care and indifference were 0.74 and 0.74, respectively, indicating acceptable internal consistency.

#### 2.2.5. Chen Internet Addiction Scale (CIAS)

The 26-item CIAS [41] was used to assess adolescents’ core symptoms of Internet addiction and related problems of Internet addiction in the previous month. Two sample questions are: “My child feel restless and irritable when the Internet is disconnected or unavailable” and “My child makes it a habit to sleep less so that more time can be spent online.” Parents rated each item on a 4-point Likert scale ranging from 1 (very unlikely) to 4 (very likely). The mean scores were calculated. A higher score indicates more symptoms of Internet addiction and related problems. The Cronbach’s α in the present study was 0.97, indicating strong internal consistency.

### 2.3. Statistical Analysis

Data analysis was performed using SPSS software v25 (SPSS, IBM, Chicago, IL, USA). Descriptive statistics and Pearson correlation analyses were first conducted to provide information about the sample and relations among the key variables. Hierarchical linear regression was performed in four steps to test the hypotheses of examining the multidimensional correlates of parental self-efficacy in managing adolescent Internet use and to ascertain the significance of potential predictors and the percentage of variance explained. Four sets of independent variables included demographics (adolescent age and sex and parental education), mental health (adolescent ADHD and ODD symptoms and parents’ depressive symptoms), parenting behavior (parental care and indifference), and adolescent behavior (Internet addiction). Adjusted R^2^ was included in determining the percentage of variance explained by each step and the overall model.

### 2.4. Ethics

The research proposal was approved by the Institutional Review Boards of Kaohsiung Medical University Hospital (KMUHIRB-20130131). The purpose and processes of the study, privacy and confidentiality of data, and freedom of participation were fully explained to all potential participants. All participants signed written informed consent.

## 3. Results

The results of descriptive statistics and correlation coefficients for study variables are presented in Table 1. All the bivariate correlations between independent variables and the dependent variable were statistically significant, except parents’ depressive symptoms. The results indicated that adolescent age, ADHD symptoms, ODD symptoms, parental indifference, and adolescent Internet addiction were negatively correlated with parental self-efficacy in managing adolescent Internet use (PSMIS), whereas parental education and parental care were positively correlated. Using the guideline of the absolute value of r [42], the correlations between Internet addiction and PSMIS were strong, and correlations between other predictors (e.g., parental care, parental indifference, ODD symptoms) and PSMIS were moderate, and the correlations between other subtypes of ADHD (e.g., H/I symptoms and inattention symptoms) and PSMIS were weak. Additional correlation analysis for other demographic variables found that parents’ sex and age and marital status were not correlated with parental self-efficacy. Using the diagnostic cut-off point of the CIAS [43], the prevalence rate of Internet addiction in adolescents with ADHD was 31.6%. Compared to the previous research in Taiwan using the same instrument, the prevalence rate of Internet addiction among adolescents with ADHD in the present study was two times higher than those among adolescents in the general public (15.8%) [7].

The results of hierarchical linear regression revealed that adolescents’ age, ODD symptoms, and Internet addiction were negatively associated with parental self-efficacy in managing adolescent Internet use, whereas parental care was positively associated (Table 2). Specifically, the first step of the hierarchical linear regression model examining the demographic variables revealed that less-educated parents and older age of adolescents were associated with lower parental self-efficacy in managing adolescent Internet use. Mental health factors were further selected into step 2, which indicated that after controlling for the effects of demographic factors, higher levels of ODD symptoms in adolescents were significantly associated with lower parental self-efficacy. Parenting behaviors were further added into step 3, which indicated that after controlling for the effects of demographic and mental health factors, higher parental care was significantly associated with higher parental self-efficacy. Finally, adolescent Internet addiction was added into step 4, which indicated that after controlling for the effects of demographic and mental health and parenting factors, higher levels of adolescent Internet addiction were associated with lower parental self-efficacy.

In the final model, parents of older adolescents with higher levels of ODD symptoms and Internet addiction were more likely to report lower parental self-efficacy levels in managing adolescent Internet use. By contrast, parents with higher levels of parental care were more likely to report higher levels of parental self-efficacy in managing adolescent Internet use. We did not observe significant associations between parental self-efficacy and variables such as adolescents’ sex, parents’ years of education, hyperactivity/impulsivity, inattention, parents’ depressive symptoms, and parental indifference. The final model explained 42% of the variance.

## 4. Discussion

The present study examined the extent to which particular dimensions of mental health, parenting, and adolescent behaviors affected parental self-efficacy in managing Internet use among adolescents with ADHD. The results of the final model indicated that high levels of adolescent ODD symptoms and Internet addiction were associated with decreased levels of parental self-efficacy in managing adolescent Internet use, after controlling for other covariates. By contrast, a high level of parental care was associated with an increased parental self-efficacy level.

In the mental health dimension, only adolescent ODD symptoms and not core ADHD symptoms and parental depressive symptoms were significantly associated with parental self-efficacy in managing adolescents’ Internet use. Although both ADHD and ODD of adolescents may increase parental stress and challenges in raising and disciplining their children, ODD seems to have a stronger influence than hyperactivity/impulsivity and inattention on parental self-efficacy in managing adolescent Internet use. ADHD adolescents with ODD often display a pattern of hostile, disobedient, and defiant behaviors directed at parents or other authority figures [44,45]. These behavioral symptoms of adolescents along with their irritable mood often make parents feel frustrated, overwhelmed, and tired, and they then start to doubt their ability and competence of being a good parent [46,47]. This may in turn reduce parental self-efficacy and affect their parenting behavior and parent–child relationship.

Although parental depressive symptoms may reduce parental self-efficacy [4], we did not observe a significant association between the two in our sample after controlling for demographic covariates and children’s ADHD symptoms. The study of [18] examined parents’ self-efficacy in the general parenting role and parenting in a US low-income minority population and found a negative association between parental depression and parental self-efficacy. The present study examined specific parental self-efficacy in managing ADHD adolescents’ Internet use. The difference in the contents of parental self-efficacy may partially account for the discrepancy of the results of this and previous studies.

In the parenting dimension, we found that parental care was positively associated with parental self-efficacy in managing adolescent Internet use, consistent with a previous study in Korea [23]. How parents treated their children was related to how they perceived themselves as parents. For example, mothers who perceive high parental self-efficacy are more likely to display warm and caring parenting behaviors toward their children [24]. Similarly, the results of the present study have shown that parents who display warm and caring attitudes toward their children were more likely to perceive high levels of parental self-efficacy and feel competent in rearing their children in the digital era. Caring is a positive and desirable parenting behavior. When parents did their best to display caring and warm behavior to their children, they considered that they did their part by demonstrating socially desirable behaviors as parents for the child’s best interests and perceived themselves as good parents no matter the challenges or stresses in raising, disciplining, and interacting with their children.

In the adolescent behavior dimension, adolescent Internet addiction was negatively associated with parental self-efficacy in managing adolescent Internet use. Given the interactive nature of the parent–child relationship, parents and children may influence each other’s behaviors. According to transactional theory [48], a set of ongoing interactions between a child and parent results in the modification of each other’s behaviors. Although parenting behaviors influence children’s externalizing behaviors and risks of Internet addiction [31,49], children may also play an active role in influencing parental behaviors and self-efficacy. The results confirmed our hypothesis that the level of adolescent Internet addiction was negatively associated with parental self-efficacy in managing adolescent Internet use. When adolescents are addicted to the Internet, they are less likely to form a good relationship with parents and more likely to have frequent conflicts with them about the use of the Internet, which in turn contributes to lower parental self-efficacy.

### 4.1. Practical and Clinical Implications

This study seems to be the first, to our knowledge, to address parental self-efficacy, specifically in managing adolescent Internet use. In the digital age, parents are concerned about adolescents’ problematic Internet use and online safety, and managing it has been a major challenge in parenting, focusing on how parents perceive themselves as competent in handling these new parenting challenges in which they may not have any background knowledge and reference point linked to their generations. This study extends the literature and examines how multidimensional aspects of mental health, parenting, and child addictive behavior on the Internet contribute to parental efficacy in managing Internet use among adolescents with ADHD.

This study has several implications for practice. On the basis of the findings, ODD symptoms were the strongest and only significant mental health factor contributing to parental self-efficacy. The findings highlight the need for family-based interventions that address the associations between adolescents’ ODD symptoms and parental self-efficacy in the negative cognitive process, which in turn may influence parenting behaviors and their interactions with their children. Moreover, when planning a family-based intervention in adolescents with ADHD, health care professionals need to be aware of adolescent Internet addiction, as this may intensify the challenges of parenting and heavily reduce parental self-efficacy. In addition to ADHD treatment, thus, it is crucial to recognize signs and symptoms of Internet addiction, empower and provide parents with parenting resources and tools, and suggest measures to improve parent–child relationships and parental self-efficacy. Given the importance of contextual factors, intervention may include parenting training that addresses the positive effects of parental care, which Asian parents do not often overtly express and the power and effectiveness of which they may not recognize [50], and promotes a positive family environment to enhance parental self-efficacy in Asian societies. Our findings can aid in the development of sophisticated prevention and intervention programs to enhance parental self-efficacy in managing Internet use among adolescents with ADHD.

### 4.2. Limitations

Although the present study provides insights into the multidimensional correlates of parental self-efficacy in managing adolescent Internet use, the study has some limitations. First, because self-report questionnaires were used to collect data one time from one source—parents of adolescents with ADHD—the results might be influenced by shared method variance (common method bias). Future studies could collect data from both parents and adolescents to reduce the shared method variance. Second, the cross-sectional design precluded the assessment of causal relationships between variables. Given the interactive nature of parent–child relationships, future studies could further investigate bidirectional relationships between parent and child behaviors through a longitudinal design. Third, the present study was conducted in a clinical sample, and the results may not be generalized to the general public. Further research should test the model in other populations.

## 5. Conclusions

In summary, the present study addressed the importance of parental self-efficacy in managing adolescent Internet use in the digital age and identified multidimensional correlates at both the individual and contextual levels. We found that adolescents’ ODD symptoms and Internet addiction were negatively associated with parental efficacy, whereas parental care was positively associated with parental self-efficacy in managing adolescent Internet use after accounting for several covariates. Our findings are consistent with transactional theory [4], which holds that children may play an active role in influencing parental behavior and self-efficacy, whereas parenting behaviors may also influence children’s externalizing behaviors and risks of Internet addiction [31]. Finally, the present study highlights the need for family-based interventions among adolescents with ADHD and the importance of recognizing signs and symptoms of Internet addiction, empowering and providing parents with parenting resources and tools, and improving parental self-efficacy.

## Figures and Tables

**Table 1 ijerph-17-05768-t001:** Bivariate correlations, means, and standard deviations for key variables in the models.

Variables	1	2	3	4	5	6	7	8	9	10
1. PSMIS	--									
2. Child age	−0.28 **	--								
3. Parent’s years of education	0.17 *	−0.07	--							
4. Child ODD symptoms	−0.32 **	0.13	−0.14 *	--						
5. Child H/I symptoms	−0.15 *	−0.04	−0.10	0.65 **	--					
6. Child inattention symptoms	−0.19 **	0.05	0.02	0.54 **	0.63 **	--				
7. Parent’s depressive symptoms	−0.13	0.12	−0.16 *	0.33 **	0.24 **	0.24 **	--			
8. Parental care	0.35 **	−0.06	0.11	−0.09	−0.13	−0.06	−0.01	--		
9. Parental indifference	−0.30 **	−0.03	−0.17 *	0.07	0.05	0.02	0.08	−0.55 **	--	
10. Internet addiction	−0.55 **	0.20 **	−0.17 *	0.31 **	0.18 **	0.25 **	0.24 **	−0.20 **	0.16 *	--
*Scale Range*	0–6	11–18	6–22	0–3	0–3	0–3	0–3	1–4	1–4	1–4
*Mean*	4.36	13.73	13.48	1.23	0.97	1.41	0.74	3.21	1.99	2.12
*SD*	1.02	1.82	2.69	0.72	0.67	0.67	0.37	0.44	0.50	0.66

Notes. PSMIS: Parental self-efficacy to manage adolescents’ Internet use scale; ADHD: attention-deficit/hyperactivity disorder; ODD: oppositional defiant disorder; H/I: hyperactivity/impulsivity; *SD:* standard deviation. * *p* < 0.05. ** *p* < 0.01.

**Table 2 ijerph-17-05768-t002:** Summary of hierarchical regression analysis for variables predicting parental self-efficacy to manage children’s Internet use.

	Model 1			Model 2			Model 3			Model 4		
Variables	*B*	*SE*	*β*	*B*	*SE*	*β*	*B*	*SE*	*β*	*B*	*SE*	*β*
**Step 1: Demographic**												
Child age	−0.17	0.04	−0.29 ***	−0.14	0.04	−0.25 ***	−0.13	0.03	−0.24 ***	−0.10	0.03	−0.17 **
Child sex	−0.13	0.19	−0.04	−0.09	0.19	−0.03	0.05	0.18	0.02	0.20	0.16	0.07
Parent’s years of education	0.06	0.02	0.16 *	0.05	0.02	0.13 *	0.03	0.02	0.08	0.01	0.02	0.03
**Step 2: Mental Health**												
Child ODD symptoms				−0.42	0.12	−0.30 **	−0.41	0.11	−0.29 ***	−0.29	0.10	−0.21 **
Child H/I symptoms				0.15	0.14	0.10	0.20	0.13	0.13	0.12	0.12	0.08
Child inattention symptoms				−0.10	0.12	−0.07	−0.11	0.12	−0.07	−0.02	0.10	−0.01
Parent’s depressive symptom				0.04	0.18	0.01	0.04	0.17	0.02	0.20	0.15	0.07
**Step 3: Parenting Behavior**												
Parental care							0.59	0.16	0.26 ***	0.47	0.15	0.20 **
Parental indifference							−0.28	0.14	−0.14 *	−0.23	0.13	−0.11
**Step 4: Child Behavior**												
Internet addiction										−0.65	0.09	−0.42 ***
*Adjusted R* ^2^		0.10			0.16			0.27			0.42	

Notes. ADHD: attention-deficit/hyperactivity disorder; ODD: oppositional defiant disorder; H/I: hyperactivity/impulsivity. * *p* < 0.05; ** *p* < 0.01; *** *p* < 0.001.

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
