# Peer review of "Multidimensional Correlates of Parental Self-Efficacy in Managing Adolescent Internet Use among Parents of Adolescents with Attention-Deficit/Hyperactivity Disorder"

_ijerph, 2020, doi:10.3390/ijerph17165768_

Round 1

Reviewer 1 Report

The manuscript titled "Multidimensional Correlates of Parental Self-Efficacy in Managing Adolescent Internet Use Among Parents of Adolescents With Attention-Deficit/Hyperactivity Disorder" investigates the relationship between parenting behaviors and child internet usage.

Introduction:

Line 40: the word efficacy is bigger than the others, please adjust.

Authors introduce the definition and importance os parental selfi-efficacy in managing their children, not only for an overall healthy childhoof as also to manage internet addiction problems.

Line 55: authors declare that "ADHD is a neurodevelopmental disorder common in children and adolescents." what do they mean by common? How are the number in Taiwan, in the world for this disorder? Please, present some statistics.

Almost after every small section in introduction authors delcare "we hypothesized that higher levels of child Internet addiction are associated  with lower parental self-efficacy levels in managing adolescent Internet use.", this is the hypothesis of the study, it must not be repeated every section. Authors can just present the theoretical relationships and definitions and resume their goal, otherwise is too repetitive.

Methods:

There is only one group of study: adolescents with ADHD, how do they believe to compare this group to a "healthier" one? Authors failed in collecting and analyzing data for a control group, results can not be comparative.

Results:

Authors report significant correlations but they didn't present a reference or the values that they consider as, for example, moderate or strong correlation, statistical significant in this case doesn't mean that the correlation is strong .

Authors report the results of the regression analysis as only "associated", but they should better exactly describe how the variables influence the parental self-efficacy. Presenting the estimated values for each significant variable is a way to better explain this.

Final comments:

Unfortunately authors lack on the most interesting comparison for parental self-efficacy: how this changes (or not) if parents from children without ADHD would be included.

Author Response

Introduction:

Line 40: the word efficacy is bigger than the others, please adjust.

[Response]: Done.

Line 55: authors declare that "ADHD is a neurodevelopmental disorder common in children and adolescents." what do they mean by common? How are the number in Taiwan, in the world for this disorder? Please, present some statistics.

[Response]: This sentence has been changed to “Attention-deficit/hyperactivity disorder (ADHD) is one of the most prevalent DSM-5 mental disorders among children and adolescents in Taiwan (8.7%) [42]”

Almost after every small section in introduction authors declare "we hypothesized that higher levels of child Internet addiction are associated  with lower parental self-efficacy levels in managing adolescent Internet use.", this is the hypothesis of the study, it must not be repeated every section. Authors can just present the theoretical relationships and definitions and resume their goal, otherwise is too repetitive.

[Response]: These sentences have been removed at the end of each section, and only presented at the end of Introduction.

Methods:

There is only one group of study: adolescents with ADHD, how do they believe to compare this group to a "healthier" one? Authors failed in collecting and analyzing data for a control group, results cannot be comparative.

[Response]: The study is not an experimental study but rather a cross-sectional study among parents of adolescents with ADHD. As stated in the limitations, the present study was conducted in a clinical sample, and the results may not be generalized to the general public or “healthier” one.

Results:

Authors report significant correlations but they didn't present a reference or the values that they consider as, for example, moderate or strong correlation, statistical significant in this case doesn't mean that the correlation is strong.

[Response]: We added sentences to illustrate the strength of the correlations in the first paragraph of Result section.

Authors report the results of the regression analysis as only "associated", but they should better exactly describe how the variables influence the parental self-efficacy. Presenting the estimated values for each significant variable is a way to better explain this.

[Response]: The estimate values/coefficients were presented in Table 2 and interpreted in the Result section. Because this is a cross-sectional research design, we can only use the term “associated with” to interpret the estimated values/coefficients in the results but cannot use the term “influence” or “cause”, unless it is an experimental design or longitudinal study. This is discussed in the limitations.

Final comments:

Unfortunately authors lack on the most interesting comparison for parental self-efficacy: how this changes (or not) if parents from children without ADHD would be included.

[Response]: Thanks for the valuable feedback. As one of the limitations, “the present study was conducted in a clinical sample, and the results may not be generalized to the general public. Further research should test the model in other populations.”

Reviewer 2 Report

Dear Authors,

I have now taken the time to review your study.

I have uploaded my comments in a separate PDF files that you will find attached.

I will be happy to check your revised manuscript, if need be.

Author Response

This study by Hsieh et al aims at assessing variables influencing Internet use in adolescents with ADHD, in relation to parenting skills. From my point of view, the study is well conducted.

The manuscript is well written and easy to understand. However, a few points will need further explanations, especially in the method section. These are detailed below, concerning both main and minor points.

I Main concerns:

1- Throughout the manuscript, author present ‘internet addiction’ and ‘internet use.’ Authors should explicitly state the difference(s) between these two. If I understood correctly, authors measured internet ‘addiction’ through the CIAS scale (Chen Internet Addiction scale). However, nowhere do they present the results of scoring in the ADHD adolescents. Therefore, how was internet addiction determinded? Did authors use a cut-off criteria in the scoring? Besides, what was the distribution of internet addiction amongst their patients? Finally, authors need to carefully choose between writing ‘internet use’ and ‘internet addiction’, throughout their study. Besides, ‘internet use’ should also be placed in the keywords.

[Response]: The outcome variable in this study is “parental self-efficacy in managing adolescents’ Internet use”, and “Internet addiction” is one of the predictors. We add a few sentences in the 1.4. in Introduction Section. In addition, the Internet addition variable was treated as a continuous variable in this study. We conduct an additional analysis to present the prevalence rate using the cut-off point of the CIAS, suggested by Chen et al., (2005). Using the diagnostic cut-off point of the CIAS, the prevalence rate of parent-report Internet addiction among adolescents with ADHD was 31.6%. This result is added in the first paragraph of the Result section. Moreover, “Internet use” is placed in the key words.

2- In the abstract (line 24) and in the method section (line 144), authors mention that 237 parents were included/recruited (n=237). From the statistics they present (83% of women responders, line 145), one would assume that only one out of the 2 parents were recruited (since

82% of participants were in a two-parent family, lines 145-146). Authors should therefore explicitly state if these 237 parents included only one parent for each ADHD adolescent, or not.

[Response]: Only one parent per household answered the questionnaire. This sentence is added in the Participant section.

3-  In  their method  section,  authors  mention  several  times  the  α and  their respective values. Authors should indicate the meaning of such values, as well as the formula(s) for calculations.

[Response]: Cronbach’s α was calculated by the SPSS statistics software to indicate the reliability (internal consistency) of the scale. Higher score means better internal consistency. In general, a score of more than 0.7 is acceptable, between 0.7 and 0.8 is good, and more than 0.9 is excellent. We add the meanings of these values in the Instruments section.

4- In Table 1, authors present 10 different columns (numbered 1 to 10). What do they represent?

[Response]: Table 1 is a correlation table. These numbers (1 to 10) represent the 10 variables listed in the first column on the left.

5- Authors should state (within an ethical statement section) how consent was obtained, who could access the results of their study, how data protection was enforced, etc... These are standard and mandatory sections required for any study involving humans.

[Response]: The information in these sections are mandatory for any study involving human, and it has been submitted and approved by the Institutional Review Boards of Kaohsiung Medical University Hospital. We have included the ethics statements in the Ethic section.

6- To put their data into perspective, authors should briefly present statistics on Internet use/addiction in the general population.

[Response]: In the Result section, we add statistics on Internet addiction in the general population. “Using the diagnostic cut-off point of the CIAS [44], the prevalence rate of Internet addiction in adolescents with ADHD was 31.6%. Comparing to the previous research in Taiwan using the same instrument, the prevalence rate of Internet addiction among adolescents with ADHD in the present study was two times higher than those among adolescents in general public (15.8%) [45].”

II Minor concerns:

1- Line 40, ‘efficacy’ is written with an odd font.

 [Response]: Done.

2- Line 48, ‘lack of reference(s) for ‘spend many hours online.”

[Response]: Reference is added.

3- Line 62. What is the difference between the SNAP-IV and the DSM-V? Both are mentioned in the manuscript at different locations. Readers would assume that the SNAP-IV assesses the severity of ADHD symptoms, while the DSM-V is rather used for the diagnostic per se of ADHD. Is that correct?

[Response]: Yes. The 26-item SNAP-IV is a self-report scale answered by parents to access the severities of their children’s ADHD and ODD symptoms. DSM-V is the Diagnostic and Statistical Manual of Mental Disorders as an important source to diagnose ADHD and ODD (and other mental disorders) in clinics.  

4- Line 69, please rephrase the sentence ’feel themselves’, as this makes no grammatical sense.

[Response]: We changed the sentence as the following: “Parents of children with significant ODD symptoms tend to view their children as more difficult to manage and view themselves as incompetent and overwhelmed.”

5- Line 82, please check if you mean ‘paternal self efficacy’ or ‘parental self efficacy.”

[Response]: We mean “paternal self-efficacy.” We changed the word “paternal” to “fathers’” and add a transition word in the next sentence to reduce the confusion.

6- Lines 118 and 125. Again, authors should check if they mean ‘internet addiction’ or ‘internet use’. This is a redundant comment with concern I-1.

[Response]: We add a word “problematic” before “Internet use”. Again, the outcome variable is parental self-efficacy in managing adolescent Internet use. And, “Internet addiction” is one of the predictors in the study.

7- Lines 157, 161, 162, 168, 180 and 185. Again, what is the ‘Cronbach’s α’? Please also refer to comment I-3.

[Response]: Cronbach’s α indicates the reliability (internal consistency) of the scale. Higher score means better internal consistency.

8- Line 181, section 2.2.5, authors mention the results from the CIAS scale. Could they provide some examples of the questions used within this test? Moreover, ‘internet addiction’ or ‘internet use’? Please also refer to comment I-1.

[Response]: Chen Internet Addiction Scale (CIAS) was used in this study to measure Internet addiction. And, Internet addiction is one of the predictors in this study. The CIAS assess adolescents’ core symptoms of Internet addiction and related problems of Internet addiction in the previous month. We add sample questions in this section.

9- Lines 239-249, in the discussion section. This whole paragraph lacks reference(s).

[Response]: In the second paragraph of the Discussion section, we add several references.

10- Lines 258-264, within the discussion section. Do authors mention here the results of the present study? If not, references are lacking. Please amend accordingly.

[Response]: Yes. In the fourth paragraph of the Discussion section, we discussed the results of the present study. Additional reference was added and a few sentences were added.

11- Line 310-311. What does ‘reorganize signs of internet addiction’ mean? Please rephrase, as this sentence makes no sense.

[Response]: It is “recognize”, not “reorganize.” And, signs of Internet addiction are similar to symptoms of Internet addiction. People usually try to understand and recognize signs and symptoms of Internet addition or other disorders or disease (e.g. depression, diabetes) to prevent or reduce their negative effects/consequences. To reduce the confusion, we add a word “symptoms” and the sentence becomes “it is crucial to recognize signs and symptoms of Internet addiction…”

12- Lines 312-316. Here, to who do authors compare Asian parents? Again, these sentences lack reference(s). When making such bold statements, please include foundations (such as references).

[Response]: The present study was conducted in a sample of Taiwanese parents. Thus, it is important to consider the contextual factors such as Asian culture and related parenting beliefs when discussing the results and implications in Asian society. Reference was added.

13-Line 319. This paragraph is mentioned as ‘4.3’ bit paragraph ‘4.2’ is missing. Please amend/check.

[Response]: The subheading should be “4.2. Limitations”, not “4.3.”. It was an error.

Round 2

Reviewer 1 Report

Authors have clearly responded to the questions and changed the manuscript to a better understanding.

It is acceptable in this format.